# Convergence of Biofabrication Technologies and Cell Therapies for Wound Healing

**DOI:** 10.3390/pharmaceutics14122749

**Published:** 2022-12-08

**Authors:** Motaharesadat Hosseini, Andrew J. Dalley, Abbas Shafiee

**Affiliations:** 1School of Mechanical, Medical and Process Engineering, Faculty of Engineering, Queensland University of Technology, Brisbane, QLD 4059, Australia; 2ARC Industrial Transformation Training Centre for Multiscale 3D Imaging, Modelling and Manufacturing (M3D), Queensland University of Technology, Brisbane, QLD 4059, Australia; 3Herston Biofabrication Institute, Metro North Hospital and Health Service, Brisbane, QLD 4029, Australia; 4Royal Brisbane and Women’s Hospital, Metro North Hospital and Health Service, Brisbane, QLD 4029, Australia; 5Frazer Institute, Translational Research Institute, The University of Queensland, Brisbane, QLD 4102, Australia

**Keywords:** biomaterial, cell therapy, matrix, regeneration, skin, stem cell

## Abstract

Background: Cell therapy holds great promise for cutaneous wound treatment but presents practical and clinical challenges, mainly related to the lack of a supportive and inductive microenvironment for cells after transplantation. Main: This review delineates the challenges and opportunities in cell therapies for acute and chronic wounds and highlights the contribution of biofabricated matrices to skin reconstruction. The complexity of the wound healing process necessitates the development of matrices with properties comparable to the extracellular matrix in the skin for their structure and composition. Over recent years, emerging biofabrication technologies have shown a capacity for creating complex matrices. In cell therapy, multifunctional material-based matrices have benefits in enhancing cell retention and survival, reducing healing time, and preventing infection and cell transplant rejection. Additionally, they can improve the efficacy of cell therapy, owing to their potential to modulate cell behaviors and regulate spatiotemporal patterns of wound healing. Conclusion: The ongoing development of biofabrication technologies promises to deliver material-based matrices that are rich in supportive, phenotype patterning cell niches and are robust enough to provide physical protection for the cells during implantation.

## 1. Introduction

Consisting of three main layers, the epidermis, the dermis, and the hypodermis, the skin is the largest organ of the body that accommodates a highly complementary network of diverse cell types to maintain its physiology [1]. Large, non-healing skin wounds, including acute and chronic wounds, impose a considerable humanistic and economic burden worldwide. Although the pathology of wound healing is well-documented [2], the available wound therapies still fall short of ideal, emphasizing the need for more research into enhanced treatments [2,3]. The incidence of skin injuries is rising as a sequel to diabetes, obesity, an aging population, and a more sedentary lifestyle [4]. Therefore, there is a growing trend in patients’ demand for minimally invasive and non-invasive wound care services, which have in recent years shifted the attention of wound research towards cell therapy. 

Considerable progresses have been accomplished in cell biology fields, and the existing evidence has revealed the effectiveness of cell therapy for pathologic wounds [5]. Transplantation of keratinocytes, fibroblasts, platelets, and more recently, stem cells (SCs) can promote wound healing through de-novo synthesis, secretion, and release of a wide range of cell signaling molecules such as growth factors (GFs) and cytokines [6,7]. Although direct cell therapy can be effective in many clinical situations, its widespread application in clinical practice is restricted by the significant challenges of building a supportive and inductive environment for transplanted cells. Engineering the cellular microenvironment requires the use of complex matrices with the ability to mimic the native extracellular matrix (ECM) and to prompt the regeneration of injured tissues [8,9]. Current advancements in biofabrication technologies enable the manufacturing of multifunctional matrices with biophysical and biochemical characteristics that partially mimic the ECM present in human skin [10,11]. Without going into the detail of cell biology and wound healing pathology, this review presents a synopsis of some of the exciting opportunities and practical challenges associated with using cell therapy to promote cutaneous wound healing, with an emphasis on the practical advantages that engineered matrices can bring to regenerative cell-based therapy. 

## 2. Significant Cell Populations for Regenerative Skin Wound Therapies 

In this section, the main cellular components in cell therapies for non-healing skin wounds are highlighted, with a focus on the challenges associated with their clinical applications. 

### 2.1. Keratinocytes

The cultivation of human keratinocytes was first introduced by Rheinwald and Green in 1975 [12]. Using this original technique, complete sheets of keratinocytes can be cultured and enzymatically released from their supporting substrate as cultured epithelial autograft (CEA) and used to surface optimally prepared partial-thickness cutaneous wounds. Six years after the pioneering work of Rheinwald and Green, O’Connor et al. reported, for the first time, the transplant of cultured autologous epithelium onto two patients with full-thickness burn wounds [13]. The cultured epithelia developed an epidermal structure similar to the split-thickness skin grafts and survived for about eight months. Importantly, given sufficient time, large numbers of CEA can be manufactured from a comparatively small initial biopsy of a patient’s skin, making CEA engraftment suitable for large wounds [13]. Decades later, cultured autologous keratinocytes have been applied for the treatment of numerous different skin wound types, and their ability to promote wound healing has been widely acknowledged [14]. 

Some studies have even explored the application of allogeneic keratinocytes in the treatment of deep partial-thickness burns and other wounds [15,16]. The beneficial effects of keratinocytes on wound healing are related mainly to their physical presence in re-epithelializing the wound. In addition, keratinocytes secrete and deposit numerous GFs, such as vascular endothelial growth factor (VEGF), platelet-derived growth factor (PDGF), basic fibroblast GF, transforming growth factor (TGF)-α, TGF-β, and cytokines such as interleukin (IL)-1, -6, -8, and -10 directly into the wound site [17,18]. Keratinocytes alone, however, are unable to produce the large, structural, and voluminous components of the ECM that comprise the dermis [19]; therefore, in isolation they are not appropriate for the re-epithelialization of full-thickness wounds. The biggest disadvantage of therapeutic autologous keratinocytes culture for the re-epithelialization of extensive cutaneous wounds is the time taken to grow sufficient CEA sheets. Using the techniques of Rheinwald and Green and starting with an initial skin biopsy of 7 cm^2^–10 cm^2^, it takes a minimum of three weeks to produce sufficient sheets to cover three quarters (75%) of an adult human’s body surface area. Other reports have suggested disadvantages including the inconsistent graft take rates, poor long-term durability, infections, mechanical shear, scarring, and high production cost [20]; however, such criticisms often reflect deficiencies in how the CEA sheets have been used clinically, rather than the fundamental properties of CEA. When used as part of a considered wound management plan in appropriate surgical settings on optimally prepared wound beds where effective post engraftment wound management practices are implemented, CEA provides a highly reliable, therapeutic adjunct for timely epithelial closure of partial-thickness wounds.

### 2.2. Fibroblasts

Autologous cultured fibroblasts are another cell therapy candidate that has been utilized for burn wounds [10,11], gingival tissue repair [21,22], and the prevention of acne scarring [23]. When transplanted effectively, fibroblasts can release numerous GFs and cytokines into the wound bed that contribute to endogenous cell proliferation, stimulate angiogenesis, and modulate the immune responses [24,25]. Of note, fibroblasts can secrete various ECM proteins such as collagens and proteoglycans and improve the wound healing rate [26]. It has been suggested that the injection of fibroblasts into the wound site could heal chronic wounds rapidly and with no adverse clinical or immunopathologic effects [27,28]. Allogenic fibroblasts can be prepared and stored in advance, but are known to be highly immunogenic and may be the source of cross-infections. Autologous fibroblasts, on the other hand, present little additional risk of infection or immune response, but require a long cultivation period which is a major practical challenge that limits their clinical applications. To date, most of the studies that have utilized fibroblasts or their derivatives to promote wound healing have used allogenic cryopreserved products that comprise poor therapeutic benefits. Some of these reports have cited the harsh recipient tissue microenvironment, which is deficient in oxygen and nutriments, as the underlying cause of poor engraftment efficiency when introducing cryopreserved fibroblasts into wounds. In addition, it has been suggested that cryopreservation causes functional impairments in fibroblasts, leading to decreased viability, impaired protein synthesis, and reduced angiogenic factor secretion [29,30,31].

### 2.3. Platelets 

Platelets contribute to normal wound healing and have been used experimentally to deliver GFs into the wound sites [32,33] such as PDGFs (PDGF-AA, PDGF-AB, and PDGF-BB), TGF-β (TGF-β1 and TGF-β2), VEGF, and epidermal growth factor (EGF). It has been suggested that platelet-derived GFs could effectively restore impaired GF activity in the wounds that are deficient in leading cells [34]. Studies have reported positive results of wound healing upon treatment with autologous platelets. The majority of patients whose platelet treatment continued for more than three months showed complete healing and reported fewer adverse events [35,36]. The major disadvantage of using autologous platelets therapeutically centers around the large-volume blood withdrawals required to isolate the platelets which may give rise to severe adverse effects, such as hemodynamic instability in patients with chronic wounds, bleeding disorders, or anemic conditions [37]. 

### 2.4. Stem/Progenitor Cell Therapies 

Differentiated cells have limited self-renewal capacity; therefore, the use of stem/progenitor cells, especially mesenchymal stem cells (MSCs), has received a great deal of interest for wound healing. MSCs present considerable immunomodulatory potential [38], and after transplantation into the wound sites, MSCs secrete ECM molecules that activate re-epithelialization, improve wound closure, and induce angiogenesis [39]. MSCs also contribute to the recruitment of several immune cell types via the release of cytokines [40,41], and can induce the differentiation of multiple progenitor cells and prompt the release of bioactive factors that support wound healing [42,43].

Considering the stem/progenitor cells’ capability to enhance wound healing, many clinical studies have been conducted to use these cells for the treatment of non-healing wounds (Table 1). Different types of MSCs, including bone marrow-derived MSCs, adipose-derived MSCs, placental-derived MSCs, and umbilical cord-derived MSCs, have been considered for cell therapy of chronic wounds. For example, Lee et al. recruited 15 patients with critical limb ischemia to be treated with multiple intramuscular adipose tissue-derived MSCs injections [44]. No complications were reported during follow-ups at a mean time of 6 months. There was clinical improvement in 66.7% of the patients. Although five patients underwent minor amputation, the amputation sites indicated complete wound healing. Additionally, the cell transplantation resulted in collateral vessel development across the affected arteries [44]. In another study, three patients with sacral pressure sore were treated with CD34+ cells isolated from bone marrow (NCT00535548) [45]. The treatment improved granulation tissue formation and wound contraction, leading to around a 50% decrease in the volume of the pressure sore on the treated side, as opposed to a 40% decrease on the control side [45].

Although highly promising (Table 1), therapeutic stem/progenitor cell engraftment has encountered several practical challenges. Obtaining high-quality progenitors for therapeutic engraftment is a slow and laborious process that requires a high level of capital investment and technical expertise making it a complex therapeutic option [62]. In addition, the potential for some progenitors to differentiate into divergent phenotypic lineages adds to the need for long-term post-engraftment surveillance to ensure that only beneficial effects have been delivered by the progenitors.

Compounding this situation, co-morbidities such as diabetes, vascular disease, and aging have the potential to drive progenitors mal-differentiation through pathological changes in the wound microenvironment [63]. Regarding post-engraftment surveillance, cutaneous wounds are particularly suitable for exploring the therapeutic potential of progenitors, since the skin is an easily accessible tissue from which problematic areas can be rapidly identified and readily excised. Progenitor cell therapy for cutaneous wounds is a developing field; the beneficial outcomes of which have so far been limited and inconsistent. There is a paucity of data for post-engraftment progenitor cells take rates, with reports of engraftment failure arising from bacterial colonization and unsuitable wound conditions [19]. 

Indeed, the injected cells have low retention rates in the transplantation sites in parts because of washout by blood flow [64,65]. In addition, ischemia and inflammation within the wound microenvironment can jeopardize the survival and proliferation of administered cells and may cause cell death in vivo [66]. Despite the technical challenges and practical setbacks that stem/progenitor therapy has encountered, it is still a very exciting field with the potential to deliver great clinical outcomes when it has been mastered.

## 3. Contribution of Matrices to the Improvement of Cell Therapy

### 3.1. Biofabrication Technologies 

Current applications of cell therapies for non-healing skin wounds are facing several challenges including poor cell retention and survival, time-consuming cell culture, prolonged duration of cell therapy, infection, and cell rejection after transplantation. A growing amount of research has examined the attributes of cell matrices to address such practical and clinical challenges. 

In regenerative medicine and tissue engineering, biofabrication technologies refer to automatic fabrication methods whereby biologically functional constructs with structural organization are produced by bioprinting or bioassembly using living and non-living building blocks, such as cells, cell aggregates (e.g., spheroids), biomolecules, and materials [67]. The first approach, bioprinting, involves computer-aided design and manufacturing of two (2D)- or three (3D)-dimensional architectures with spatially arranged structures, while bioassembly deals with the development of 2D or 3D hierarchical structures through the automated assembly processes of preformed living units [68]. Different techniques have been developed to incorporate cells into matrices for wound healing (Table 2). 

As shown in Table 2, various biomaterials, natural and synthetic, have been used to improve cell therapy in the wound context. These biomaterials require an appropriate printability, provide geometrical accuracy, and have adequate biophysical properties (e.g., mechanical strength, biocompatibility and biodegradability) [81]. Natural materials have been found to enable cell adhesion, resemble the ECM of the native skin tissue, and have suitable biocompatibility [10,11]. However, natural biomaterials suffer from low mechanical strength, which, in turn, causes a lot of problems during processing and material manipulation. Thus, it is critical to make sure that the use of biofabrication technologies does not involve any risk to the viability, metabolic activity and functionality of cells. In this regard, both material properties (e.g., type of solvent, rheology, or concentration) and processing parameters (e.g., pressure, voltage, temperature, or feed rate) play key roles in shielding the cells against potential stresses related to the biofabrication process, and accordingly in promoting cellular behaviors [81]. On the other hand, synthetic biomaterials are consummate candidates due to their biochemical and biomechanical properties [82]. For example, poly(vinyl alcohol) (PVA) is a water-soluble, organic polymer with excellent biocompatibility, biodegradability and processability [83]. Xu et al. have studied the application of PVA in cell electrospinning for wound healing [69]. Since the synthetic materials lack appropriate cell adhesion features and sufficient bioactivity, some suggest semi-synthetic materials such as gelatin-methacryloyl (GelMA) that allow for the utilization of the biological signals innate to gelatin, as well as taking advantage of mechanical properties [73].

The contribution of biofabrication technologies to cell therapy has been supported by an increasing body of literature. For example, biofabricated matrices control cell growth, migration, infiltration and differentiation through the sufficient recapitulation of the microenvironment physiologically relevant to the skin cells [78], and strong interaction with physicochemical properties, including porosity [75,76], topography [69,79] and crosslink density [72]. More details are found in previous reviews [10,11]. In the following sections, the key roles of biofabricated products to develop a supportive and inductive matrix for cell therapies are discussed.

### 3.2. Matrices as Supportive Carriers 

Effective cell delivery is pivotal to the outcome of cell-based therapies and can be achieved in several ways (Figure 1). Cells administered systemically via infusion or locally via injection may quickly succumb to apoptosis/cell death, or exhibit poor viability resulting in short-term engraftment [84,85,86,87]. Wu et al. reported that engraftment rates for bone marrow-derived MSCs decreased from 28% in the first week to 2.5% four weeks following injection [88]. The undesirable consequences of syringe injection may result from exposing cells to high shear stresses, the lack of ECM for cells to bind with and form a cellular network, leakage or mechanical washout of cells from the wound site, or host pathological responses causing inflammation and the production of reactive oxygen species (ROS) [89]. Various natural and synthetic biomaterials have been developed to establish a suitable niche for cells to adhere and grow.

Matrices made of biopolymers such as collagen and fibrin can provide appropriate ECM constituents with healing effects [10,11]. Razavi and Thakor showed that poly(dimethylsiloxane) could promote the viability of seeded adipose-derived MSCs [90]. There are many synthetic polymers, including poly(ethylene glycol) [91], poly(lactic-co-glycolic acid) (PLGA) [92] and poly(methyl methacrylate) [93] which can support cell growth for skin regeneration. The porous microstructure of these synthetic polymers can maximize the benefits of biomaterials, which can reduce healing times from a range of 10–28 days [94,95] to a range of 7–14 days [96,97]. Additionally, the porous matrices provide an appropriate reservoir for incorporating GFs and controlling cell growth, and the presence of interconnected pores facilitates cell–cell and cell–matrix communications. Accordingly, the porous microstructure of matrices improves cellular attachment, proliferation and differentiation in ways that resemble an in vivo microenvironment [98]. Moreover, the porous structure of biopolymer matrices accelerates the infiltration of endogenous cells from the surrounding tissue into the matrix [99], while allowing the therapeutic delivery of exogenous cells.

To establish an optimal healing environment, wounds must be covered in ways that maintain the beneficial moisture content in the wound vicinity. Hydrogels are a class of biomaterials with the ability to support the survival and growth of transplanted cells and boost the healing process. Cells encapsulated within the hydrogels can detach from it, and migrate into a wound bed and ultimately participate in the healing process [100]. Importantly, when used as carrier substrates, hydrogels protect cells from the mechanical forces that they are exposed to during their delivery [101]. Additionally, in specific applications, the properties of hydrogels can be manipulated to protect cells from immune rejection [89]. 

Surface modifications of hydrogels using cell-binding peptides, antioxidant ligands, or the inclusion of nanoparticles can impart protective effects to hydrogels to shield cells against ROS, hypoxia, and inadequate nutrient supply [102]. Cell encapsulation within the matrices can prevent cell leakage from injection sites and has notable benefits to the survival and retention of transplanted cells [103]. Engineered biomaterials with controlled biodegradability can provide physical support to cells, enhance cell engraftment and promote the restoration of hierarchical tissue architecture. Controlled rate polymer biodegradation is useful in promoting cell integration into host tissues by encouraging delivered cells to migrate and deposit ECM as the biomaterial is resorbed [104]. The combination of biodegradability and the porous microstructure of biomaterials provides encapsulated cells with protective support when they are being delivered and while higher cell numbers could be seeded on the biomaterials, and also provide sufficient space for cells to migrate and proliferate, which can ultimately increase engraftment rates [105]. 

Traditional and advanced biofabrication strategies have been used to modulate the microstructure of biomaterials [106,107,108]. One such method that is highly applicable to skin tissue engineering is 3D (bio)printing. It is described as the biomimetic layer-by-layer deposition of materials, non-living biologicals and living cells within a 3D pattern in a controllable manner [109]. This technology paves the way for finely creating 3D structures with complicated geometries and precisely regulated spatial positioning of cells and biomaterials [10,11]. These features can be adopted to develop tissue-engineered matrices/substitutes with diverse cellular compositions similar to human skin. For challenging wounds such as ulcers, skin substitutes can be developed using keratinocytes and fibroblasts. When pro-angiogenetic signals are impaired in wounds, the healing process is interrupted by the lack of vascularization or limited tissue growth and integration; in which, the engrafted cells fail to survive and proliferate after transplantation [110]. To induce wound healing, 3D bioprinted constructs composed of well-defined layers of human skin fibroblasts, keratinocytes and microvascular endothelial cells have been manufactured [111]. Post-engraftment immunohistochemistry results indicated the survival of the implanted cells and showed that the human cells contributed to skin repair after transplantation [111]. 

In situ 3D bioprinting is more potent than in vitro 3D bioprinting for complex wounds, where the direct de novo formation of human tissues is planned with specific anatomical features [112]. This approach capitalizes on the potential of inkjet bioprinting technology, either via hand-held or automated systems, for the direct development of cell-laden constructs. In this way, cell suspensions at a higher density can be applied to wound sites, leading to better cell-cell communications and enhanced healing. The study by Albanna et al. can be a proof-of-concept for a mobile skin bioprinting system, whereby concentrated suspensions of human dermal and epidermal cells were delivered through a fibrin–collagen bioink onto an extensive excisional full-thickness wound in vivo [79]. The printed skin substitutes contain high skin cell quantity and could retain this high cellularity up to six weeks after printing. This treatment resulted in a rapid wound closure, decreased wound contraction and accelerated re-epithelialization. The dermal structure and composition of the restored tissues compared favorably with unwounded skin, showing organized collagen fibers, mature vascular networks and proliferating epidermal cells [79]. Considering the irregularity and complexity of acute and chronic wounds, both in vitro and in situ printing techniques are viable alternatives to conventional scaffolding methods, being able to layer different cell types and materials at specific ratios according to the precise topography of wounds.

### 3.3. Matrices as Inductive Substrates to Modulate the Biophysical and Biochemical Responses

#### 3.3.1. Biophysical Cues

Biomaterials can potentially modulate the biophysical and biochemical signals that facilitate cell proliferation and differentiation, improving wound healing and skin regeneration. This section discusses the role of biophysical features of engineered matrices in modulating cell behaviors. Apart from porosity, biophysical cues can include matrix stiffness, topography and external loading (stress and strain). To migrate within the wound environment, cells alter their matrices establishing tiny channels to move through. The prospect of cell migration relies on the stiffness of the matrices [113,114]. It has been reported that a marked elevation of the matrix stiffness leads to a significant reduction in cell speed because of the high physical barriers required to be degraded to facilitate cell motility [115]. Through enhanced cellular interactions with the matrix, stiffness can accelerate the wound healing process, particularly hemostasis and proliferation [116].

Surface topographies affect matrix surface properties, including hydrophilicity, surface energy, and cellular interactions with no significant changes in the bulk properties [117,118]. By having the micro/nanopatterned topographical factors, biomaterials regulate cellular responses toward the development of a self-organized cell-derived matrix suitable for tissue repair [119,120]. The morphology of fibroblasts and epithelial cells, and the organization of the actin cytoskeleton, can conform to the topographic cues, with stiffer microgrooves aligning the cells more effectively [121]. Topographies in the form of squares (100–500 μm^2^) have been found to guide the migration of dermal fibroblasts and epidermal keratinocytes for wound healing. These microfeatures on the matrix surface could provide the adherent cells with a protective effect against shear damages [122]. Besides the intrinsic biophysical features of the matrix, external mechanical stimuli such as stress or strain can carry an impact on cellular behaviors. Stress fiber traction plays a pivotal role in mediating the interplay between extension and contraction, as two imperative factors related to the dynamical adjustment of cell adhesion and spreading [123]. For example, there are three phases for the adhesion and spreading of keratinocytes on their matrix under gradient strains, upon nondirectional dynamic mechanical stretching. By applying the 5% and 8% strain on the matrix, keratinocytes showed actin stress fibers and tight cell–cell junctions (phase 1). With increasing the tensile strain on the matrix, rising from 10% to 15%, larger-sized keratinocytes indicated stable adhesion (phase 2). When the strain level reached 22%, keratinocytes went through phase 3, experiencing forces that were too high and could destroy the stability of the stress fibers in parts. Therefore, mechanical loading led to spatiotemporal responses in keratinocytes [124], which are of utmost importance in wound healing. 

Of note, biomaterial tolerance to mechanical stresses, such as the wound contractile forces, results in a delay in wound contraction and contributes to less scarring [125]. Indeed, biomaterials act as a “contraction-blocker”, paving the way from repair to regeneration with less scarring (Figure 2) [125]. This role of biomaterials reorganizes the orientation of assemblies of myofibroblasts and fibrous collagen networks in the process of wound healing [126].

#### 3.3.2. Matrices for Efficient Delivery of Bioactive Molecules

Growth factors play an important role in supporting cell survival, proliferation, growth, and differentiation. Thus, some studies suggest the application of exogenous GFs to enhance cell behaviors [11]. The delivery of these bioactive molecules to adherent cells culminates in the activation and acceleration of tissue regeneration [76]. As exogenous GFs may be prone to degradation, biomaterials can serve as a substrate on which GFs can be conjugated according to chemical, topographical, and mechanical features [127,128]. For example, Gümüşderelioğlu et al. immobilized EGF on poly(caprolactone) (PCL)/gelatin and PCL/collagen scaffolds via the amine end-groups. Biological studies indicated the contribution of EGF immobilization to the growth, proliferation and migration of keratinocytes and fibroblasts [129,130]. In these scenarios, biomaterials are considered delivery vehicles for GFs. The convergence of drug delivery systems and biofabrication techniques enables the construction of bioactive biomaterials that act as skin substitutes, and support cellular functions and tissue formation. The incorporation of GFs into the tissue-engineered matrices is performed through five strategies: surface presentation, controlled sustained release, preprogrammed release, responsive release and gene transfection. Table 3 shows some of the studies that combine GFs with biomaterials for wound healing and skin regeneration applications.

Biomaterials with controlled release of bioactive molecules can create an appropriate microenvironment for cell delivery, enhance cell proliferation and engraftment levels, and ultimately improve the cell therapy outcomes [11,137]. Some studies added spirulina, known as a blue-green microalga, to the matrix and assessed the contribution of such a construct in cutaneous wound healing [138,139]. Steffens et al. fabricated skin substitutes using electrospinning, with or without spirulina biomass [140]. The poly(D, L-lactic acid) (PDLLA)/spirulina scaffolds were more capable of being molded with better adherence to the wound bed, as opposed to the PDLLA scaffolds. After implantation in mice, both scaffolds tolerated the mechanical stress without rupture for 14 days. The PDLLA/spirulina scaffolds showed greater potential for cell delivery as they contained a higher MSCs density in comparison with the PDLLA-alone scaffolds [140]. 

Another bioactive molecule is hematoporphyrin, which plays a key role as photosensitizer to not only produce ROS and prevent microbial growth, but also encourage cell proliferation, and control inflammatory responses and ECM remodeling [141]. Koo et al. employed the exogenous ROS-induced cell sheet stacking method to develop hematoporphyrin-incorporated polyketone films [142]. After irradiating with light, the films were removed and the cell sheets were transferred onto the fibrin gel as a cell carrier. This process continued until a multi-layered human MSCs sheet was generated in vitro. In this design, the films and fibrin gel addressed the poor localization of the implanted MSCs. The ROS/reactive nitrogen species generated from the film delivered sufficient oxygen and nutrients to MSCs. Of note, there was a difference in wound healing outcomes (i.e., wound area, thickness of epidermal layer, scar formation and production of skin appendages) between single and three-layered MSCs sheet transplantation, cell suspension injection and a non-treated control for three weeks. In a full-thickness wound model, the multi-layered MSCs sheets contributed to enhanced angiogenesis and skin regeneration in vivo [142].

For a skin tissue-engineered scaffold, it is essential to induce regeneration, deliver oxygen and nutrients and avoid infection during the healing process [110]. Prevention of wound infections is the main focus of wound care to circumvent any delay in normal wound healing. Matrices with incorporated anti-infective molecular agents such as antibiotics, antimicrobial peptides (AMPs), etc., have held great promise for the acceleration of the healing process and inhibition of wound pathogens. In this regard, Wang et al. encapsulated gentamicin in PLGA microspheres, which were subsequently added to collagen/chitosan mixtures and manufactured into a two-compartment and bi-functional scaffold [134]. The release profile of gentamicin from PLGA microspheres included three classic periods, burst release during the first two days, linear release up to the 15th day and slow release exceeding 28 days. Biological studies have exhibited improved adhesion and proliferation of mouse fibroblasts. Furthermore, the scaffolds could effectively prevent the growth of *Staphylococcus aureus* and *Serratia marcescens*, implying their optimal antibacterial activities [134].

Antimicrobial peptides are indispensable elements of innate immunity. They are short and cationic peptides that show a diversity of activities against different microorganisms [143]. Some of the AMPs target both microorganisms and human cells, in particular immune cells, and modulate immune responses [143] and the designation of host defense peptides (HDPs). For instance, Kasetty et al. incorporated the thrombin-derived HDPs, i.e., GKYGFYTHVFRLKKWIQKVI and GKYGFYTHVFRLKKWIQKVIDQFGE, on a human acellular dermis to fabricate a dermal substitute with combined antimicrobial and anti-endotoxic effects. The functionalized dermis could inhibit the activation of nuclear factor kappa B in human monocytic cells and reduce pro-inflammatory cytokine release in whole blood upon exposure to lipopolysaccharide [144].

An ideal regenerative biomaterial can support the in-growth, attachment and proliferation of endogenous and exogenous cells. To this aim, one could incorporate peptide sequences into the material so that specific binding sites facilitate cell attachment. In this regard, Glycyl-Histidyl-Lysine (GHK), as a typical matrix-derived tripeptide, has become the focus of tissue engineering research. GHK peptide plays a crucial role in modulating neo-tissue formation [145]. GHK peptide has been found to function as a cell adhesion biomolecule whereby endogenous cells can attach to the ECM and function (to migrate, proliferate, or differentiate) [145]. Arul et al. functionalized collagen films with biotinylated GHK for dermal wound healing application. The enhanced proliferation of fibroblasts and production of collagen were observed, which showed the ability of GHK peptides to attract fibroblasts to the wound site [146].

Of the defined peptides capable of promoting cell activities, RGD peptide (Arg-Gly-Asp) is the crucial modifier of scaffolds and mostly resides in collagen, gelatin, elastin, fibronectin and laminins. It forms an anchoring site for integrin receptors that augment cell adhesion and proliferation [147,148]. Dong et al. used an in situ-formed hydrogel to deliver adipose-derived MSCs into the burn wounds [149]. The hydrogel contained a hyperbranched poly(ethylene glycol) diacrylate, thiol-functionalized hyaluronic acid, and a short RGD peptide. The RGD peptide, as a cell adhesion motif, improved cell proliferation and promoted the paracrine effects of MSCs, mediating the angiogenesis and tissue remodeling processes. The hydrogel protected the transplanted MSCs against the detrimental burn wound environment. Treatment with hydrogel-MSC remarkably enhanced burn wound healing outcomes, including neovascularization, wound closure and scarring [149]. Apart from RGD, there are several different pro-adherence sequences isolated from collagen (DGEA, GFOGER, and GFPGER peptides) [147], fibronectin (RGDS, PHSRN, REDV, LDV, and KQAGDV peptides) [147,150,151] and laminins (IKVAV and YIGSR peptides) [147,152,153] that have the potential to improve cell adhesion and function. 

Together, the behavior of stem/progenitor cells can be determined by factors in their immediate surrounding environment. To maintain the cell capacity for growth, proliferation, differentiation and to accordingly establish tissue structure and function, these cells require biophysical and biochemical signals; the matrix, as the local microenvironment surrounding the cells, is a critical facet of this signaling. Besides providing shielding support, matrices can carry biochemical factors and regulate their delivery spatiotemporally in their placement. The matrices can also transfer biophysical regulatory information to the cells and modulate repair mechanisms (Figure 3).

## 4. **Stem/Progenitor Cells Seeded Matrices in Clinical Settings**

Scaffold-assisted cell transplantation has demonstrated promising outcomes in the treatment of cutaneous wounds in numerous preclinical investigations [154,155,156], which can be utilized in clinical settings. This section highlights the clinical trials where a combination of cells and matrices has been applied for wound healing.

Epidex® is composed of confluent autologous keratinocytes isolated from the hair follicle outer root sheath of scalp tissues and cultured on a silicone membrane. In a multicenter, randomized phase two study on patients with recalcitrant vascular leg ulcers, Tausche et al. compared wound healing outcomes between EpiDex (*n* = 43) and split-thickness skin grafts (*n* = 34) [157]. Both treatments were almost similar and enhanced the healing rate with complete wound closure. Of note, EpiDex supported better quality of life, defined as smaller ulcer-related pain and lower disability [157]. 

In a retrospective study on 68 patients with chronic wounds (i.e., chronic leg ulcers and sores), Ortega-Zilic et al. evaluated the contribution of EpiDex to complete wound closure [158]. According to the results, 74% of the patients showed complete wound healing, with 78% experiencing complete pain disappearance. However, 22% required antibiotic therapy for wound infection and 3% presented dermatitis (not associated with the local treatment) [158].

Another off-the-shelf product for skin wound treatment is Dermagraft^®^, consisting of 3D ECM, human dermal fibroblasts and a bioabsorbable polyglactin mesh. In a prospective, multicenter, randomized, controlled 12-week study, Hanft et al. recruited 28 patients with chronic ulcers, who were then divided into Dermagraft treatment or control groups [159]. After three months, the Dermagraft group had a considerably higher number of healed wounds than the control group (71.4% versus 14.3%), also with the treated patients reporting less infection rates at the wound sites. They also achieved higher percentages of wound closure compared with the control patients [159]. In a large clinical trial with an open-label, prospective, multicenter, randomized controlled design, Harding et al. compared Dermagraft (*n* = 186) with compression therapy (*n* = 180) in 366 patients who suffered from venous leg ulcers [160]. There was no marked difference in healing rate between the two groups by three months. Comparatively, both groups showed similar adverse events, including wound infection, cellulitis and skin ulcer. The healing rate of ulcers ≤ 12-month duration was substantially greater in the Dermagraft group than those under compression therapy [160]. 

The third skin construct, named Apligraf^®^ or Graftskin^®^, benefits from both fibroblasts and keratinocytes seeded on structural proteins. The lower dermal layer comprises bovine type 1 collagen and human fibroblasts, responsible for the secretion and release of matrix proteins. The upper layer contains human epidermal keratinocytes, which primarily form a monolayer and then become stratified. Applying a prospective, randomized, and controlled trial, Falanga and Sabolinski investigated venous leg ulcers for more than one-year duration [161]. Treatment groups were compression therapy, or its combination with Graftskin. Graftskin outweighed the control treatment group in terms of healing rate (47% versus 19%) and the median time to complete wound closure [161]. Further evaluations by this research group revealed that Graftskin was more effective than compression therapy in accomplishing complete wound closure in hard-to-heal wounds [162]. Developing an international multicenter, randomized, controlled study, Edmonds et al. reported 84 days as the median time to healing [163], which was longer than that recorded in the study by Veves et al. (65 days) [164]. In the treatment of noninfected nonischemic chronic plantar diabetic foot ulcers, the application of Graftskin was associated with some adverse effects, such as osteomyelitis and lower-limb amputations [164]. 

StrataGraft^®^ contains a dermal equivalent made from an animal-derived, murine source of type I collagen with human dermal fibroblasts, as well as a fully-stratified epidermis obtained from NIKS cells (human keratinocyte progenitor cell line) which are known as pathogen-free, long-lived, genetically-stable, human keratinocyte progenitors. Using a prospective, randomized, controlled dose-escalation trial, Centanni et al. explored the efficacy and safety of StrataGraft in 15 patients with full-thickness skin wounds [165]. StrataGraft was found to be well-tolerated and did not cause acute immunogenicity. Importantly, there was no increase in the patient’s sensitivity to immune responses against the NIKS cells, upon treatment with StrataGraft [165]. More recently, Gibson et al. have conducted a phase three, open-label, controlled, randomized, multicenter trial on StrataGraft in patients with deep partial-thickness thermal burns [166]. Results showed that 92% of the patients treated with StrataGraft experienced durable wound closure at month 3, without autografting. Although 15% of patients developed pruritis after the treatment, but there was no need for autografting in the StrataGraft-treated group and the relevant donor-site morbidities are prevented [166]. 

The studies mentioned above are proof-of-concept proving the feasibility of scaffold-assisted cell transplantation in wound healing and skin regeneration. Additionally, the use of biomaterial vehicles for the delivery of MSCs has been reported in a few human studies. Falanga et al. applied a fibrin matrix containing bone marrow derived MSCs to treat patients with acute and chronic wounds [167]. Portas et al. adopted the same strategy for cases with large burns and radiological lesions [168]. It has been found that the treatment was safe with no adverse events, enhancing the healing rate of the wounded tissues and preventing consecutive inflammatory reactions. Of note, the higher the number of MSCs applied, the smaller the size of chronic wounds was [167,168]. Besides fibrin gel, there are some research investigations in which platelet-rich plasma (PRP) gel was used for MSC delivery [169]. The PRP consists of fibrin and GFs required for the wound healing process [10,11]. Findings revealed that the healing was faster, with an invariably high magnitude [169,170].

## 5. **Current Limitations and Future Opportunities**

The development of the scaffold, able to recapitulate the microenvironments experienced by incorporated stem/progenitor cells, is the main focus of the biofabrication technologies mentioned above. However, the resultant matrices are passive per se, with no capability to monitor these cells after transplantation. Hence, future research should consider developing active matrices capable of reporting cellular bioactivity and even reacting to the surrounding stimuli effectively. Some studies suggest electronic sensors to keep track of the cell physiology on 2D planar surfaces [171,172]. Nevertheless, it is daunting to combine electrical sensing moieties with 3D matrices. Tian et al. took the first step toward the production of a bioactive 3D scaffold, by applying a hybrid of silicon-nanowire field-effect transistors with natural (collagen and alginate) or synthetic (PLGA) materials [173]. This scaffold could monitor the viabilities of neurons and cardiomyocytes over a long time, as well as the local electrical activity of cardiomyocytes [173]. 

On the other hand, the regeneration of whole skin tissue with no or less scarring requires functional vascular networks that enable the exchange between skin cells and blood for gases, nutrients, and metabolic products [110]. The majority of the available biofabricated matrices can accommodate cells only for a short time in an almost small environment. Integrating a network of microvessels, with stem/progenitor cells seeded matrices using biofabrication technologies, would support the formation of larger and thicker tissue over long periods. Additionally, scar formation can be caused by wound infection and high tension at specific anatomical sites [123,174]. Therefore, future studies should focus on the integration of vasculature and neural networks within the skin substitutes, using the biofabrication technologies to create cell-containing matrices with the capabilities of large skin tissue formation, infection prevention and tension reduction.

## 6. **Conclusions**

Bioengineered matrices with integrated biomolecules provide a desirable microenvironment for cell function, which can be tailored to provide the temporospatial cues that orchestrate tissue-specific patterns of cell differentiation. The ongoing development of biofabrication technologies promises to deliver material-based matrices that are rich in supportive, phenotype patterning cell niches and are robust enough to provide physical protection for the cells during implantation. Judicious and well-executed application of these technologies to the treatment of acute and chronic wounds is likely to overcome many of the current obstacles to efficacious cell-based therapy for these conditions that carry a high morbidity burden. 

Any vehicle used to engraft cells has the potential to impact autocrine and paracrine activity and affect cellular differentiation. In terms of cell biology, bioengineered matrices act as bioactive membranes or as artificial substitutes for the cellular niches that are normally present in unwounded tissues. When used as cell engraftment vehicles, these properties of bioengineered matrices support the functional recovery of transplanted cells and improve the structural integrity and remodeling of newly formed tissues in vivo, which can enhance therapeutic efficacy. Further therapeutic efficiency can be attained by employing matrices that improve the localization of stem/progenitor cells and reduce their washout from the implantation site to ensure that the stem/progenitor cells remain in situ for long enough to execute their therapeutic role.

The application of bioengineered matrices to cell-based therapy of cutaneous wounds benefits not only the exogenously applied therapeutic cells, but also the tissue-resident endogenous cells. The physical substrate and its functional cellular niches encourage the retention and engraftment of exogenous cells, while the combined effects of a porous bioactive matrix and the cell signaling molecules, such as GFs and cytokines, that emanate from the therapeutic cells, and support and guide endogenous (skin resident) cells to contribute more beneficially to the wound healing process. In this way, an optimally functioning cell/matrix combination has the potential to improve the final scar outcome for patients by resisting the stress and strain from local tissue contraction, and by dampening the profibrotic phenotype of endogenous cells—particularly tissue resident fibroblasts. In many ways, the incorporation of bioengineered matrices into cutaneous wound healing eases the interaction of cells with their surrounding environment, dictating cell fate decisions such as adhesion, proliferation, migration and differentiation; processes which must all be coordinated throughout wound healing for the regeneration of new stable tissues. 

The emergence of therapeutic engineered biomolecular substrates, that can be carefully paired with stem/progenitor cells, and modified into products that preferentially promote beneficial phenotypic traits—in both exogenously applied cells and tissue-resident (endogenous) cells—holds great potential to address many hitherto unmet clinical demands. Such approaches pave the way for further translational and clinical research to develop new therapeutic and preventive concepts, and will give a unique insight into cell biology, material science and their crosstalk. There is a need for ongoing research to address some of the known limitations of this evolving technology. While many of the currently used bioengineered matrices enormously enhance cell growth and proliferation, they often fail to unleash the full therapeutic potential of cells by not promoting appropriately patterned cell differentiation. The challenge now is how best to design matrices with the range of biophysical and biochemical parameters that balance the need for cell adhesion, proliferation and migration with the complimentary, yet sometimes conflicting, need for cells to commit to the pattern of terminal differentiation that is required for them to establish a hierarchical tissue architecture. There is also the need for a greater understanding of how the properties of newly developed material-based matrices, that employ biofabrication technologies, can be used to enhance the integration of neo-tissues into the body—with particular emphasis on neovascularization, wound contraction and any pro-fibrotic effects. Currently, most of the marketed skin substitutes contain cells other than stem cells; therefore, there is both need and opportunity for the development and clinical implementation of tissue-engineered skin substitutes that combine stem cells and biomaterials. 

## Figures and Tables

**Figure 1 pharmaceutics-14-02749-f001:**
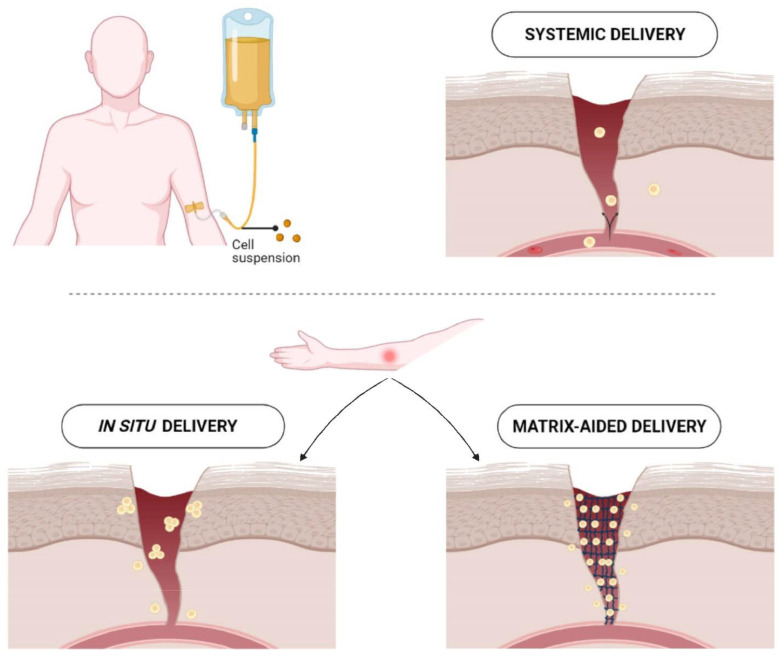
Different routes of cell delivery to cutaneous wound. In clinical interventions, cell therapy is conducted using systemic delivery, which has off-target effects. Although it affords localization of cells, in situ delivery fails to maintain therapeutic doses. Matrix-aided delivery of cells is considered a promising breakthrough for overcoming the existing challenges and providing more opportunities.

**Figure 2 pharmaceutics-14-02749-f002:**
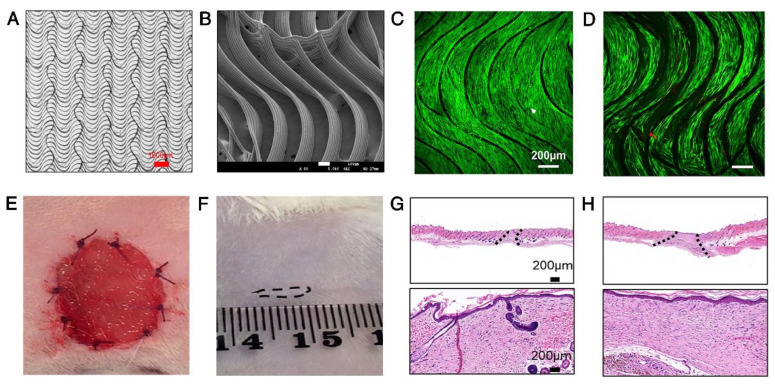
The proof-of-concept of the convergence of biomimetic matrices and stem/progenitor cell therapy. (**A**) Light microscopy and (**B**) scanning electron microscopy photographs show the microstructure of the anisotropic biomimetic medical-grade poly(caprolactone) (PCL) wound dressing. (**C**,**D**) The confocal laser scanning microscope images of fresh (**C**) and frozen (**D**) PCL wound dressings seeded with human gingival mesenchymal stem/stromal cells. Bioengineered constructs support the attachment, survival, and proliferation of cells. (**E**) The PCL wound dressings could adhere to the wound bed and be sutured to the surrounding tissues. The tissue-engineered PCL dressings were tested in an excisional wound in the rat model. After 6 weeks, the wounds are healed with reduced scarring (**F**) and complete epithelialization in fresh (**G**) and cryopreserved (**H**) tissue-engineered constructs. Adapted with permission from Elsevier, 2022 [125].

**Figure 3 pharmaceutics-14-02749-f003:**
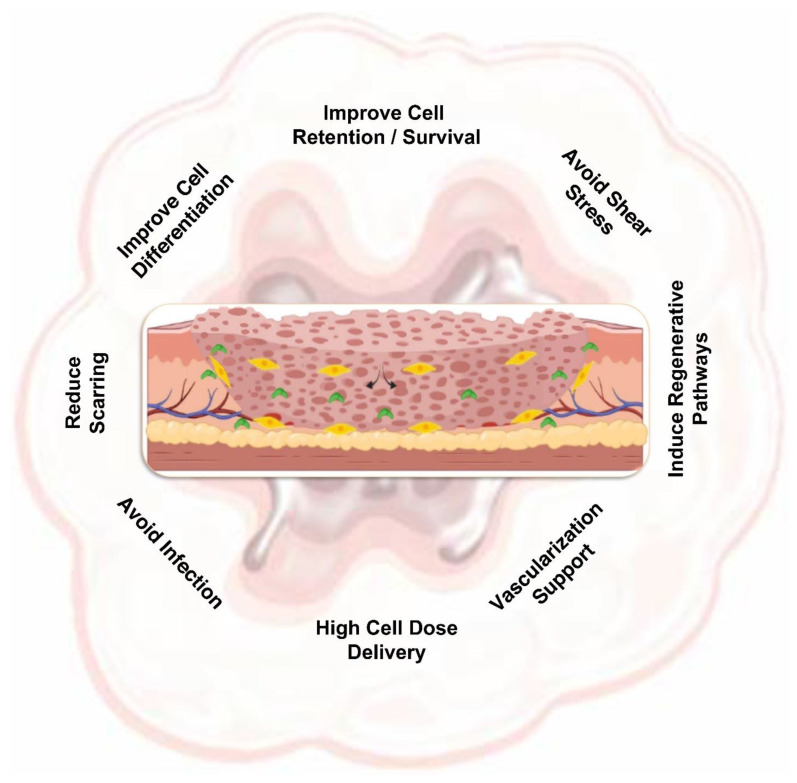
A schematic representation of the contribution of biofabricated matrices to cell therapy in cutaneous wounds. By virtue of their role as a supportive carrier, biofabricated constructs could carry huge amounts of cells and protect them in the harsh wound environment. Additionally, they incorporate bioactive features of various molecules to fight infection, induce vascularization and regeneration. Accordingly, wounds can heal without severe scars. The two arrows indicate the release of pre-loaded bioactive molecules from the matrix.

**Table 1 pharmaceutics-14-02749-t001:** Clinical studies on stem/progenitor cell-based therapy for wound healing. Abbreviations: AVLU: leg ulcers of arterial-venous, CLI: critical limb ischemia, DFU: diabetic foot ulcers, MSCs: mesenchymal stem cells, PAD: peripheral arterial disease, VLU: leg ulcers of venous.

Stem/Progenitor Cells	Treatment Group(s)	Wound Type	Remarks	Reference
Adipose-derived MSCs	Adipose tissue derived MSCs	CLI	- 66.7% of patients showed ulcer healing - The treatment showed the formation of numerous vascular collateral networks	[44] *
	1: Autologous adipose-derived stem and regenerative cells plus traditional methods and advanced dressings 2: Only traditional methods and nonadherent dressings	Chronic ulcer of lower limbs	- There was a reduction in both the diameter and depth of the ulcer - In 6 of 10 cases, there was complete healing of the ulcer	[46]
	Autologous cultured adipose-derived stroma/SCs	Non-revascularizable critical limb ischemia	- Ulcer evolution and wound healing showed improvement	[47] **
	Non-culture-expanded autologous, adipose-derived stromal vascular fraction cells	CLI	- 6 of the 10 patients with non-healing ulcers had a complete closure - There was evidence of neovascularization in 5 patients	[48] *
	Adipose-derived SCs	Hypertensive leg ulcers	- Wound surfaces constantly and significantly decreased (wound closure rate of 73.2% at month 3 and 93.1% at month 6) - Percentages of fibrin and necrosis decreased, whereas granulation tissue increased significantly - There was no recurrence	[49] *
	1: Autologous stromal vascular fraction cells plus a wound dressing 2: A standard dressing	Chronic VLU and AVLU	- All VLU patients and 4 of 9 AVLU patients showed complete epithelialization of the ulcers within 71–174 days - In 3 patients with large ulcerations on both legs, ulcerations on the non-treated, contralateral leg also epithelialized (paracrine effects seemed to stimulate the regenerative changes even at a large distance)	[50]
Bone marrow derived MSCs	1: Allogeneic bone marrow-derived MSCs 2: PlasmaLyte A	CLI	- The use of allogeneic BM-MSCs was safe in patients with CLI - All ulcers at two-year follow-up healed in group 2, whereas one patient in group 1 continued to have ulcers but with reduced size	[51]
	1: Bone marrow-derived cells 2: Autologous peripheral blood plus regular wound care treatments	Chronic lower limb wounds due to diabetes mellitus	- The average decrease in wound area at 2 (17.4% vs. 4.84%) and 12 (36.4% vs. 27.32%) weeks was higher in group 1 compared to in group 2	[52]
	1: Bone marrow MSCs 2: Bone marrow-derived mononuclear cells 3: normal saline	Diabetic critical limb ischemia	- The ulcer healing rate was significantly higher in group 1 - They reached 100% four weeks earlier than group 2 - Ulcer healing rate in group 2 was significantly higher than in group 3, which appeared at 12 weeks	[53]
	Autologous bone marrow nuclear cells	Pressure ulcers	- Pressure ulcers had fully healed after a mean time of 21 days in 86.36% of the patients - During a mean follow-up of 19 months, none of the resolved ulcers recurred	[54] *
	1: Autologous bone marrow aspirate 2: Saline dressings	Chronic wounds	- Group 1 achieved a significant reduction in the wound surface area	[55]
Progenitor cells	CD34+ cells isolated from bone marrow	Sacral pressure sore	- The treatment positively affected granulation tissue formation and wound contraction, which showed about a 50% reduction in the pressure sore volume on the treated side versus a 40% reduction on the control side	[45] **
	Genetically modified epidermal stem cells	Junctional epidermolysis bullosa	- Complete engraftment was achieved following 8 days - Transduced stem cells enabled the regeneration of epidermis	[56] **
	Genetically modified epidermal stem cells	Junctional epidermolysis bullosa	- The human epidermis is supported not by equipotent progenitors, but by long-lived stem cells with an extensive self-renewal ability so that they could generate progenitors to renew terminally differentiated keratinocytes	[57] **
Bone marrow-derived mononuclear cells	Mononuclear bone marrow cells	Chronic venous and neuro-ischemic wounds	- The treatment led to a wound size reduction, a markedly increased vascularization, and infiltration of mononuclear cells	[58] **
Placental MSCs	1: Cryopreserved human placental tissue in a human viable wound matrix plus standard compression therapy 2: Standard compression therapy	VLU	- Complete healing in 53% of the cases in group 1 - Reduction in wound size by half (80% in group 1 vs. 25% in group 2)	[59]
	Human placenta-derived mesenchymal stromal-like cells (cenplacel)	DFUs with PAD	- There was preliminary evidence of ulcer healing in seven patients (five complete; two partial) within 3 months of cenplacel treatment - Circulating endothelial cell levels (a biomarker of vascular injury in PAD) were decreased within 1 month - Cenplacel was generally safe and well-tolerated in patients with chronic DFUs and PAD	[60] *
Umbilical cord MSCs	1: Human umbilical cord MSCs plus a percutaneous angioplasty treatment 2: A percutaneous angioplasty treatment	Ulcer wounds	- 3 months after treatment, there was a significant increase in neovessels accompanied by complete or gradual ulcer healing in group 1	[61]

* This is a prospective uncontrolled study. ** This is a case report.

**Table 2 pharmaceutics-14-02749-t002:** Biofabrication technologies that are used for the delivery of cells in clinical settings. Abbreviations: 3D: three-dimensional, ECM: extracellular matrix, MSCs: mesenchymal stem/stromal cells, SCs: stem cells.

Technologies	Biofabrication Mode	Biomaterial Platforms	Biomaterial	Cell Type	Remarks	Reference
Cell electrospinning	Bioassembly	Nanofibers	Polyvinyl alcohol	Bone marrow-derived SCs	- Good infiltration and cell growth due to the even distribution of the cells throughout the fiber filaments - Acceleration of wound healing and appendage regeneration by promoting granulation tissue repair - Formation of dense and mature collagen fiber structure parallel to the epidermis	[69]
Extrusion	Bioassembly (vibrational modality)	Shell/core microcapsules	Poly(methyl-methacrylate)	Human dermal fibroblasts	- Decrease of cell viability as long as the number of microcapsules increased - After 72 h incubation, microcapsules did not interfere with cell growth - Slow cell proliferation inside the microcapsules	[70]
	Bioassembly (electrostatic droplet modality)	Microcapsules	Alginate	Human adipose-derived SCs	- Growth of the encapsulated cells in static culture by 3 weeks - Cell survival after injection into a nude mouse - Protection of the cells during injection—potential deterrent to donor cell migration	[71]
Soft lithography	Bioprinting	Microgels	Hyaluronic acid modified with photoreactive methacrylates	Fibroblasts	- Uniform distribution of the cells throughout the gel (depending on the crosslinking process) - Maintaining the cell viability (depending on the exposure time of the cells to ultra violet, photoinitiator concentration and exposure to dry air) - Cell-mediated degradation of hydrogels	[72]
Photolithography	Bioprinting	Microgels	GelMA and graphene oxide	Fibroblasts	- Support of cellular adhesion and spreading with improved viability and proliferation - Robust mechanical properties and excellent flexibility - Able to construct multilayer cell-laden hydrogels	[73]
Emulsion	Bioassembly	Hydrogels	Sodium alginate	Keratinocyte clusteroids	- Growth of the clusteroids in the hydrogels - Percolation of the clusteroids through the hydrogel and formation of an integral tissue	[74]
Microfluidics	Bioassembly	Hollow microspheres	Bacterial cellulose	Primary epidermal keratinocytes	- Increased proliferation due to the high porosity of the microsphere scaffold - Enhanced wound healing due to 3D mimicry of the native skin ECM and water retention	[75]
	Bioassembly	Microporous annealed particle gels	Multi-armed poly(ethylene)glycol-vinyl sulfone functionalized with RGD	Dermal fibroblasts, adipose-derived MSCs; bone marrow-derived MSCs	- Cell proliferation and formation of extensive 3D networks by the incorporated cells - Facilitation of the cell migration, rapid cutaneous tissue regeneration and tissue structure formation due to a stably linked interconnected network of micropores	[76]
Bioprinting	Bioprinting (inkjet modality)	Hydrogels	Fibrin and collagen	Amniotic fluid-derived SCs and bone marrow-derived MSCs	- Facilitation of quick wound and closure and angiogenesis due to delivery of secreted trophic factors - Greater re-epithelialization - Increased microvessel density - Transient integration of the cells with the surrounding tissue	[77]
	In vitro bioprinting (extrusion modality)	3D cell-printed full-thickness human skin equivalent	Decellularized ECM-based skin	Endothelial progenitor cells and adipose-derived SCs	- Sufficient recapitulation of the microenvironment physiologically relevant to the skin cells (dense and thick microstructure) - Improved epidermal organization, dermal ECM secretion and barrier function - Acceleration of wound closure, re-epithelization, neovascularization, and blood flow	[78]
	In vivo bioprinting (extrusion modality)	Two-layered skin construct (hydrogel)	Fibrinogen and collagen	Human fibroblasts and human keratinocytes	- Capable of delivering the cells to the specific target sites - Rapid wound closure, reduced contraction and accelerated re-epithelialization - Regeneration of tissues with a dermal structure and composition similar to healthy skin, with extensive collagen deposition arranged in large, organized fibers, extensive mature vascular formation and proliferating keratinocytes	[79]
	In vivo bioprinting (extrusion modality)	Skin precursor sheets	Fibrin and hyaluronic acid	Mesenchymal stem/stromal cells	- High cell viability and increased proliferation - Improved re-epithelialization, dermal cell repopulation and neovascularization	[80]

**Table 3 pharmaceutics-14-02749-t003:** The incorporation of GFs into matrices for skin wound applications. Abbreviations: 3D: three-dimensional, EGF: epithelial growth factor, HA: hyaluronic acid, IGF: insulin growth factor, PCL-b-PEG: poly(ε-caprolactone)-block-poly(ethylene glycol), PELA: poly(_DL_-lactide)-poly(ethylene glycol), PLGA: poly(lactic-co-glycolic), VEGF: vascular endothelial growth factor.

Growth Factor	Biomaterial Composition	Delivery System	Study Type	Remarks	Ref.
IGF and EGF (vitronectin: GF complexes)	HA hydrogel	Surface presentation, physical adsorption	In vitro culture of fibroblasts and keratinocytes Ex vivo model of 3D de-epidermized dermis human skin equivalent	In vitro: the combination of the complexes and HA activated the proliferation of human fibroblasts but not keratinocytes. Ex vivo: the combination improved the proliferative and differentiating layers. HA promoted absorption and transport.	[131]
EGF	Electrospun nanofibers of PCL-b-PEG	Surface presentation, chemical conjugation	In vitro culture of keratinocytes In vivo model of full-thickness diabetic wounds in mice	In vitro: the conjugation of EGF to nanofibers considerably enhanced the expression of keratinocyte-specific genes. In vivo: the conjugation led to better wound healing outcomes such as wound closure. The expression of EGF-receptor was on a significant rise.	[132]
EGF	Fibrin gel loaded within chitosan nanoparticles	Controlled sustained release	In vitro culture of fibroblasts	EGF released from the composite gel was bioactive for one week at most. It could activate the proliferation of fibroblasts.	[133]
VEGF	Two-compartment and bi-functional scaffold from chitosan/collagen-containing PLGA microspheres	Preprogrammed release	In vitro culture of fibroblasts	VEGF showed a linear release behavior over a long period (49 days). The scaffold could support cell adhesion and proliferation.	[134]
EGF	Photo-cross-linkable pluronic/chitosan hydrogel	Responsive release	In vitro culture of keratinocytes In vivo model of diabetic ulcers in mice	In vitro: EGF contributed to the retention of original phenotypes of keratinocytes. In vivo: EGF had high retention in the hydrogel at the wound site, which was in favor of the proliferation of keratinocytes. The slow release of EGF carried an effect on the keratinocytes proliferation of epidermal cells and supported wound recovery. EGF worked better in the differentiation of epidermal cells into keratinocytes, than in the acceleration of wound healing rates.	[135]
Polyplexes of bFGF	Electrospun core−sheath fibers from PELA	Gene transfection	In vitro culture of fibroblasts In vivo model of diabetic skin wounds	In vitro: bFGF improved cell proliferation. The transfection continued for over four weeks. In vivo: its release led to considerably high wound recovery with enhanced vascularization, collagen deposition and maturation, complete re-epithelialization and skin appendage restoration.	[136]

## Data Availability

Not applicable.

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
