# Peer review of "Convergence of Biofabrication Technologies and Cell Therapies for Wound Healing"

_pharmaceutics, 2022, doi:10.3390/pharmaceutics14122749_

Round 1

Reviewer 1 Report

The review is interesting and exhaustively presents the problems associated with the use of cells and scaffolds in wound healing therapies. References includes the latest scientific reports. It reads well and is very informative, I especially liked the paragraph 4. Stem/Progenitor Cells Seeded Matrices in Clinical Settings.

Very minor changes:

Page 2, line 52-55: font size

Page 20, line 430: Expand figure legend.  Arrows indicate??

Author Response

Comments and Suggestions for Authors

Reviewer #1

The review is interesting and exhaustively presents the problems associated with the use of cells and scaffolds in wound healing therapies. References includes the latest scientific reports. It reads well and is very informative, I especially liked the paragraph 4. Stem/Progenitor Cells Seeded Matrices in Clinical Settings.

Very minor changes:

Page 2, line 52-55: font size

Authors Response: We are delighted that our paper catches the reviewers’ attention and appreciate their valuable comments and have addressed them in the revision.

Page 20, line 430: Expand figure legend.  Arrows indicate??

Authors Response: More details have been added to the caption (Page 23, lines 492 – 496). The two arrows indicate the release of pre-loaded bioactive molecules from the matrix.

Reviewer 2 Report

In this article, entitled “Convergence of biofabrication technologies and stem/progenitor cell therapies for wound healing”, the authors provide comprehensive reviews of cell therapy for wound repair. There are some minor comments.

 1. The authors need to explain what the “Biofabrication technologies” are. Although the authors show several examples, it would be beneficial to provide a separate paragraph and a new figure to explain "Biofabrication".

2. The authors keep mentioning "Stem/progenitor cell therapy", but they only deal with the MSCs. Therefore, the authors need to change “Stem/progenitor cells” to “MSCs” for clarification.

In addition, progenitors of MSCs usually differentiate into mesenchymal lineage cells, such as bone, cartilage, and adipose cells. So, it does not make sense how progenitors of MSCs contribute the skin wound repair. Therefore, it still needs to be clarified 1) what exactly MSC progenitors contribute to wound healing in addition to the stem cells and 2) whether stem cells and progenitors have different roles. If stem cells and progenitor cells do not have significantly different roles, the authors should mention only stem cells, not progenitor cells.

3. In the introduction, the authors said, “the available wound therapies are not effectual.” However, it seems there are so many options for cell transplantation based on the authors’ explanation, such as CEA, fibroblasts, platelet, and stem cell transplantations. Why do you think the current therapies are not effective? The authors need to explain more details.

4. The authors did not cite the references well. For example,  they mentioned keratinocytes (Lines 79-82, page2) and fibroblasts (lines 98-100, page2) express several factors without references. The authors should carefully check the citation.

5. "Keratinocyte" and "Fibroblast" are just short single words. The authors do not need to use abbreviations (Kc, Fb).

Author Response

Comments and Suggestions for Authors

Reviewer #2

In this article, entitled “Convergence of biofabrication technologies and stem/progenitor cell therapies for wound healing”, the authors provide comprehensive reviews of cell therapy for wound repair. There are some minor comments.

Authors Response: We do appreciate reviewer # 2 comments on our manuscript and are grateful for his/her constructive suggestions which have been applied in the revisions. 

  1. The authors need to explain what the “Biofabrication technologies” are. Although the authors show several examples, it would be beneficial to provide a separate paragraph and a new figure to explain "Biofabrication".

 Authors Response: The first paragraph in section 3 (Page 9, Lines 208 – 216) deals with biofabrication technologies, which are automatic fabrication methods whereby biologically functional constructs with the structural organization are produced by bioprinting or bioassembly using living and non-living building blocks. We, hereby, emphasized on those methods that have been used for improving cell delivery for therapeutic purposes. In Table 2, we have highlighted the key biofabrication technologies that have been applied for cell delivery.

  1. The authors keep mentioning "Stem/progenitor cell therapy", but they only deal with the MSCs. Therefore, the authors need to change “Stem/progenitor cells” to “MSCs” for clarification.

In addition, progenitors of MSCs usually differentiate into mesenchymal lineage cells, such as bone, cartilage, and adipose cells. So, it does not make sense how progenitors of MSCs contribute the skin wound repair. Therefore, it still needs to be clarified 1) what exactly MSC progenitors contribute to wound healing in addition to the stem cells and 2) whether stem cells and progenitors have different roles. If stem cells and progenitor cells do not have significantly different roles, the authors should mention only stem cells, not progenitor cells.

Authors Response: the examples reported in Table 1 are the cells used for wound healing clinically. We added two more examples of progenitors in this table. In these studies, progenitors (epidermal stem cells) were used for cases with junctional epidermolysis bullosa (Refs # 56 and 57).

By term stem cells we are pointing out the cells that reside in a specialized somatic microenvironment (niche) and undergo asymmetric cell divisions, can self-renew and at the same time give rise to the tissue-specific progenitors. The progenitors (immediate descendants of stem cells) multiply in large numbers, differentiate into tissue-specific cells and thereby maintain homeostasis. The normal body stem cells are expected to get mobilized and bring about regeneration. However, controversies around the exact definition remain and the concept is still evolving. Therefore, the “Stem/Progenitor cells” is a general term that have been widely used in the previous studies 1-3.

Following the Reviewer #2 comment, the contribution of MSCs to wound healing was highlighted in the main text (Pages 5 and 6, Lines 152 – 156). These cells present considerable immunomodulatory potential 4, and after transplantation into the wound sites, MSCs secrete ECM molecules activate re-epithelialization, improve wound closure, and induce angiogenesis 5. MSCs contribute to the recruitment of several immune cell types via the release of cytokines 6,7, and can induce the differentiation of multiple progenitor cells and prompt the release of bioactive factors that support wound healing 8,9.

Progenitors like Keratinocytes have been traditionally considered inert constituents of the multilayered epidermis. The keratinocyte is now recognized as an active player in epidermal renewal with key functions in the skin's immune defense. Under homeostatic conditions, keratinocyte progenitor cells are believed to divide symmetrically or asymmetrically, that is they continue to proliferate or go on to terminally differentiate and build up the overlaying epidermis. Therefore, it would better keep both stem cells and progenitors as cell populations for cell therapy in wound healing. Following the Reviewer #2 comment, we also slightly revised the manuscript title.

  1. In the introduction, the authors said, “the available wound therapies are not effectual.” However, it seems there are so many options for cell transplantation based on the authors’ explanation, such as CEA, fibroblasts, platelet, and stem cell transplantations. Why do you think the current therapies are not effective? The authors need to explain more details.

 Authors Response: Following this reviewer’s comment we have revised this section (Page 2, Lines 50-51). These therapies are limited in their applications. For example, keratinocytes (epidermal stem cells or cultured epithelial autograft) alone, however, are unable to produce the large, structural, and voluminous components of the ECM that comprise the dermis and in isolation, therefore, are not appropriate for the re-epithelialization of full-thickness wounds. As for more details on the limitations of common cell-based therapies in wound healing, section 2 addressed them elaborately.

  1. The authors did not cite the references well. For example, they mentioned keratinocytes (Lines 79-82, page2) and fibroblasts (lines 98-100, page2) express several factors without references. The authors should carefully check the citation.

Authors Response: Thanks for highlighting this point. The supportive literature for our statements has been provided in the revisions (Ref #17, 18, 24, and 25).

  1. "Keratinocyte" and "Fibroblast" are just short single words. The authors do not need to use abbreviations (Kc, Fb).

Authors Response: The terms Kc and Fb have been replaced with Keratinocyte and Fibroblast throughout the manuscript.

  1. Huang Y-Z, Gou M, Da L-C, Zhang W-Q, Xie H-Q. Mesenchymal Stem Cells for Chronic Wound Healing: Current Status of Preclinical and Clinical Studies. Tissue Engineering Part B: Reviews. 2020;26(6):555-570.
  2. Wu Y, Wang J, Scott PG, Tredget EE. Bone marrow-derived stem cells in wound healing: a review. Wound Repair and Regeneration. 2007/09/01 2007;15(s1):S18-S26.
  3. Shin L, Peterson DA. Human Mesenchymal Stem Cell Grafts Enhance Normal and Impaired Wound Healing by Recruiting Existing Endogenous Tissue Stem/Progenitor Cells. Stem Cells Translational Medicine. 2012;2(1):33-42.
  4. Nagamura-Inoue T, He H. Umbilical cord-derived mesenchymal stem cells: Their advantages and potential clinical utility. World J Stem Cells. Apr 26 2014;6(2):195-202.
  5. da Silva LP, Reis RL, Correlo VM, Marques AP. Hydrogel-Based Strategies to Advance Therapies for Chronic Skin Wounds. Annu Rev Biomed Eng. Jun 4 2019;21:145-169.
  6. Isakson M, de Blacam C, Whelan D, McArdle A, Clover AJ. Mesenchymal Stem Cells and Cutaneous Wound Healing: Current Evidence and Future Potential. Stem Cells Int. 2015;2015:831095.
  7. Kosaric N, Kiwanuka H, Gurtner GC. Stem cell therapies for wound healing. Expert Opin Biol Ther. Jun 2019;19(6):575-585.
  8. Dabiri G, Heiner D, Falanga V. The emerging use of bone marrow-derived mesenchymal stem cells in the treatment of human chronic wounds. Expert Opin Emerg Drugs. Dec 2013;18(4):405-419.
  9. Hu MS, Borrelli MR, Lorenz HP, Longaker MT, Wan DC. Mesenchymal Stromal Cells and Cutaneous Wound Healing: A Comprehensive Review of the Background, Role, and Therapeutic Potential. Stem Cells Int. 2018;2018:6901983.
